# Infectious Microorganisms Seen as Etiologic Agents in Parkinson’s Disease

**DOI:** 10.3390/life13030805

**Published:** 2023-03-16

**Authors:** Stuparu Alina Zorina, Sanda Jurja, Mihaela Mehedinti, Ana-Maria Stoica, Dana Simona Chita, Stuparu Alexandru Floris, Any Axelerad

**Affiliations:** 1Department of Neurology, ‘St. Andrew’ County Clinical Emergency Hospital of Constanta, 900591 Constanta, Romania; zorina.stroe@yahoo.com (S.A.Z.);; 2Department of Neurology, General Medicine Faculty, ‘Ovidius’ University, 900470 Constanta, Romania; 3Department of Ophthalmology, ‘St. Andrew’ County Clinical Emergency Hospital of Constanta, 900591 Constanta, Romania; 4Department of Ophthalmology, General Medicine Faculty, ‘Ovidius’ University, 900470 Constanta, Romania; 5Department of Morphological and Functional Science, University of Medicine and Pharmacy, “Dunarea de Jos”, 800017 Galati, Romania; 6Department of Neurology, Faculty of General Medicine and Pharmacy, “Vasile Goldis” Western University of Arad, 310045 Arad, Romania; 7Department of Orthopedy and Traumatology, ‘St. Andrew’ County Clinical Emergency Hospital of Constanta, 900591 Constanta, Romania

**Keywords:** Parkinson’s disease, virus, bacteria, pathogens, neuroinflammation

## Abstract

Infections represent a possible risk factor for parkinsonism and Parkinson’s disease (PD) based on information from epidemiology and fundamental science. The risk is unclear for the majority of agents. Moreover, the latency between infection and PD seems to be very varied and often lengthy. In this review, the evidence supporting the potential involvement of infectious microorganisms in the development of Parkinson’s disease is examined. Consequently, it is crucial to determine the cause and give additional treatment accordingly. Infection is an intriguing suggestion regarding the cause of Parkinson’s disease. These findings demonstrate that persistent infection with viral and bacterial microorganisms might be a cause of Parkinson’s disease. As an initiating factor, infection may generate a spectrum of gut microbiota dysbiosis, engagement of glial tissues, neuroinflammation, and alpha-synuclein accumulation, all of which may trigger and worsen the onset in Parkinson’s disease also contribute to its progression. Still uncertain is the primary etiology of PD with infection. The possible pathophysiology of PD infection remains a matter of debate. Furthermore, additional study is required to determine if PD patients develop the disease due to infectious microorganisms or solely since they are more sensitive to infectious causes.

## 1. Introduction

In Parkinson’s disease, which is a neurodegenerative disorder, the loss of dopaminergic neurons in the substantia nigra compacta causes motor symptoms [1]. Mitochondrial dysfunction and inflammation are believed to have a significant part in the death of dopaminergic neurons despite the lack of clarity around the underlying causes [1,2].

Parkinson’s disease is a prevalent neurodegenerative disorder in the elderly, defined by resting tremor, bradykinesia, stiffness, postural instability, and walking problems [1].

Based on the finding of Lewy bodies in the vagus nuclei before their appearance in the substantia nigra, Braak et al. [3] proposed the retrograde transit of a pathogen from the gastrointestinal system to the brain. In confirming this theory, the same group documented the concurrent incidence of aggregates of alpha-synuclein in the nucleoplasm, comparable with the ones which characterize Lewy bodies inside the brain and in stomach neurons of both Meissner’s and Auerbach’s plexus. All studied patients had inclusions in the dorsal motor nucleus of the vagus and enteric nervous system; however, only patients with more advanced disease exhibited inclusions in the substantia nigra and brain [4].

Others have suggested a link with current or previous Helicobacter infection [5,6,7]; while clearance of this bacterium has been observed to ameliorate symptoms, this could be due to increased dopaminergic absorption instead of a direct influence on the illness process itself [8].

Infection is becoming acknowledged as a risk factor for Parkinson’s disease because it may produce persistent inflammation of microglia and so promote the development of PD [9]. Despite much having been discovered about Parkinson’s disease since its original description by James Parkinson, the circumstances that cause neuronal death and the mechanisms that drive its relentless development are still unknown.

Several hypotheses have been advanced on the likely origin of PD, including both environmental and genetic factors. Epidemiologic studies have linked a higher risk of developing Parkinson’s disease to a variety of employment contexts, particularly in education and the medical field [10,11]. It has not been determined whether frequent exposure to viral or other illnesses connects educators and medical professionals to the other jobs correlated with a higher chance of developing Parkinson’s disease; notwithstanding, it has been proposed that this may be the case. While it is recognized that victims of von Economo’s encephalitis may acquire postencephalitic parkinsonism years or decades later, no infectious agent has been clearly linked to the onset of Parkinson’s disease [12].

Chronic degradation of nigrostriatal dopaminergic neurons, resulting in striatal dopamine deficiency, was already identified as the primary defect that causes the apparition of Parkinson’s disease’s basic movement characteristics. It has been evidenced that the neuronal destruction in Parkinson’s disease is not restricted to the dopaminergic system of the nigrostriatum but involves additional dopamine networks within the brain, additional neurotransmitter systems, and the enteric nervous system [13]. Nevertheless, the cause of this persistent neuronal damage has yet to be identified. Several hypotheses involving oxidative stress, excitotoxicity, and proteasomal instability have been proposed [12]. Inflammatory CNS pathways have also been postulated [12].

## 2. Materials and Methods

Our research used the scientific electronic databases PubMed, Google Scholar, Web of Science, and Science Direct. Between 1980 and 2023, English papers containing the terms “Parkinson’s disease”, “etiology”, and “pathogenesis” in conjunction with “infectious”, “viral”, and “bacterial” were recognized as being significant. The selection of 49 items was determined by database queries. Recent research and case report sources were manually inserted into the search. The headings of the papers were examined for accuracy, and duplicates were removed. Furthermore, 79 articles were found; 23 were eliminated after full-text screening, and 7 were eliminated due to duplication.

The literature addressing the microbiological genesis of Parkinson’s disease fulfilled the eligibility conditions. The authors personally verified the accuracy of the reference records of the selected literature to validate the presence of pertinent information on the probable involvement of infectious agents in the genesis of Parkinson’s disease. Articles were considered for inclusion if they fulfilled specific criteria: (1) they were published in English, and (2) they were original research articles, reviews, or case reports. Studies that exhibited the following criteria were excluded: (1) citations and patents, (2) articles published in a language other than English, and (3) abstracts devoid of data.

## 3. Neuroinflammation

### 3.1. Inflammation in the Brain

Set expectations were postulated that the brain is an immunologically shielded organ, isolated from systemic infections by the blood–brain barrier (BBB) and incapable of mounting a substantial immune reaction by itself [7]. To the contrary, the situation is just the opposite. The central nervous system is completely able to set up both a coordinated initial reaction as well as the innate immune system, including a response to a bacterial infection or other trauma [9]. This reaction arises in brain regions (circumventricular organs) lacking the blood–brain barrier and act as immune system shields to the central nervous system (CNS) [9]. The actual reaction expands due to those same locations to engage brain parenchyma microglia [14]. This early immune reaction might evolve through an adaptive immune reaction intended to give lasting protection [14]. In the body’s peripheral regions, macrophages, neutrophils, dendritic cells, and natural killer cells are the main mediators of the innate immune response, whereas the adaptive immune response is marked by lymphocytes, which are subdivided into B cells, which generate the immune system’s humoral reaction, and T lymphocytes, which are responsible for cell-mediated immunological activity [14]. Glial cells, especially astrocytes and microglia, perform a dominating but maybe not exclusive function in the CNS [15]. Despite these distinctions, the CNS can generate an active immune response, not just in reaction to acute assaults from outside sources as well as to mechanisms that seem to develop inside the central nervous system itself [15].

This ability of glial cells to produce simultaneously harmful and beneficial agents is an essential element of the immunological response mediated by glial cells in the CNS [16,17,18]. In recent years, it has become apparent that disruptions in this fine balance among neuronal nourishment and neuronal elimination could be the origin of neurodegenerative disorders such as Parkinson’s disease [16,17,18].

### 3.2. Astrocytes in Neuroinflammation

Astrocytes contribute to the homeostasis of the neuronal extracellular environment in the normal brain [19]. They serve an important function in regulating extracellular potassium content and in the absorption of synaptic transmitters as well as the preservation of the blood–brain barrier [20]. Triggered by damage or other insult, astrocytes might rescue surviving adjacent neurons by cleaning (through glutathione peroxidase) oxygen free radicals generated by dying neurons, eliminating the surplus of extracellular glutamate and releasing neurotrophic factors [21].

### 3.3. The Role of Microglia in the Inflammation of the CNS

Less than 14 percent of the CNS’ glial cells consist of microglia [15,16]. The distribution of microglia is not uniform across the CNS; their concentration was found to be especially high in the substantia nigra [22,23]. Microglia are in a state of rest and have a ramified, highly branching shape under normal circumstances [22,23]. Similar to astrocytes, microglia are triggered by perturbations in their surroundings and acquire an ameboid, macrophage-like morphology when triggered [24]. They subsequently produce proinflammatory molecules, including reactive oxygen species, reactive nitrogen species, inflammatory prostaglandins, and cytokines, including tumor necrosis factor alpha and interleukin-1beta [25,26]. In acquiring these properties, microglia may convert into phagocytic cells [27], and their activities facilitate the clearance of dead cells as well as other cell debris after damage, which are essential tasks in damage repair. Activated microglia use the complement system [28], also acquiring the capacity of delivering the antigen for T cells to perform their phagocytic function [29]. In the context of CNS inflammation, activating products of the complement system have been detected, suggesting that the complement system may be triggered by substances besides antibodies [30].

### 3.4. Inflammation of the Central Nervous System in Parkinson’s Disease

Typically, inflammatory processes that are built into the body reactions are self-limiting, diminishing as the immunological danger disappears and damage management and repair are accomplished. Under certain circumstances inside the CNS, nevertheless, the neuroinflammatory process is considered to have a self-existence and stimulation of astrocytes and microglia is maintained, presumably indeterminately, having devastating results since allegedly healthy neurons that could be mistaken for uninvolved observers in the activating area are inevitably targeted and eliminated according to what is already understood as “collateral damage” in the defense department. In Alzheimer’s disease [31] and Parkinson’s disease [29], as well as in persons with parkinsonism produced by the toxin 1-methyl-4-phenyl-1, 2, 3, 6-tetrahydropyridine (MPTP), persistent glial activation has been reported [32]. This may be the result of a persistent onslaught by a pathogen that has not yet been discovered, or it may be the result of a latent, self-perpetuating immune response that continues to thrive after the initial shock wears off.

Insufficient evidence exists to conclude oligodendrocytes are involved in the pathology of Parkinson’s disease. Moreover, astrocytes seem to have a small function. There is insufficient proof of notable astrocytic activation in the substantia nigra of people with Parkinson’s disease [33]. Myelo-peroxidase, an oxidant-producing enzyme that might harm dopamine neurons, is released by astrocytes, and it has been suggested that astrocytes could control microglial multiplication by releasing the cytokine granulocyte macrophage colony-stimulating factor (GM-CSF) [34]. By contrast, the marked and considerable increase in microglial activation was already consistently seen in PD animal models and Parkinson’s disease postmortem investigations or MPTP-induced parkinsonism [35,36,37]. This stimulation and expansion of microglia is known as the neuroinflammatory characteristic of Parkinson’s disease.

## 4. Parkinson’s Disease and Microglia: A Look at Their Involvement

Expansion of HLA-DR+ (human leukocyte antigen—DR isotype) reactive microglia in the substantia nigra of postmortem cases of PD suggests a potential role for microglial activation in PD [38]. Other researchers have elucidated and broadened the concept of neuroinflammation in Parkinson’s disease. TNF-alpha (tumor necrosis factor alpha), IL-1, IL-2, IL-4, IL-6, epidermal growth factor (EGF), converting growth factor alpha (TGF-alpha), TGF-1, and TGF-2 have been recognized at elevated levels in nigrostriatal areas of the CNS and ventricular cerebrospinal fluid of Parkinson’s disease patients, supposedly emerging from activated microglia [37,39,40]. Neuronal damage may need prolonged exposure to proinflammatory cytokines such as IL-1, according to one theory [41]. On the cell bodies and processes of nigrostriatal neurons exist receptors for proinflammatory cytokines, including TNF-alpha, which probably renders the neurons that are vulnerable to the devastating impacts of cytokines produced by stimulated microglia [36]. This receptor modulation may begin or activate a caspase cascade, culminating in the neuron’s apoptotic death [35,36].

In response to cytokines that assault nigrostriatal neurons, microglial activity might potentially harm and kill neurons through additional methods [40]. Postmortem analysis of Parkinson’s disease brains has shown indications of oxidative stress [40,41,42]. Activated microglia produce nitric oxide (NO) and superoxide, which may mix to create peroxynitrite, a highly reactive and toxic oxidant that damages and kills cells by interacting with DNA, proteins, and lipids [40,41,42]. Furthermore, NO generated by active microglia may traverse cell membranes and enter dopaminergic neurons in order to communicate with superoxide generated inside the neuron and cause oxidative damage that inevitably results in neuronal death [43]. This has been established in MPTP-induced PD animal models and is postulated to happen in PD [43].

Some researchers have hypothesized that the neuro-inflammatory mechanism in Parkinson’s disease might not be formed only in the pathways of neurons and glial cells but may potentially include immunological cells in the periphery and perhaps even brain capillaries [44,45]. Upon that premise of the discovery of interferon-gamma–positive cells in the substantia nigra of PD patients, it has been postulated that T cells invade the brain parenchyma from the circulation [46]. Unknown is the significance of these cells, although their presence may indicate that the blood–brain barrier is actually altered or compromised in PD [47].

## 5. Infectious Risk Variables

Multiple sources suggest the notion of an infectious cause or contributor to Parkinson’s disease. Early findings of clusters of patients afflicted by parkinsonism after infectious outbreaks are an example of presumed viral infections that are considered to cause parkinsonism-inducing chronic neurological disease and are consequently associated with outbreaks (Figure 1).

### 5.1. Parkinson’s Disease and Influenza

During the 1900s influenza pandemic, the fact that encephalitis lethargica frequently led to parkinsonism indicated a probable viral origin for PD [48,49,50]. Nonetheless, since that period, clinical and neuropathologic criteria have distinguished postencephalitic parkinsonism from normal idiopathic PD [48,49,50]. Despite the fact that several subsequent investigations have failed to uncover an infectious agent in PD, a plethora of studies remains to suggest that infection might have a role in the etiology of PD [48,49,50]. In addition, the increased risk of developing PD is not due to chemical exposure but rather to contact with an infectious pathogen.

The correlation between the frequency of influenza episodes and the chance of having parkinsonism suggests that influenza-correlated brain damage could be an accumulating inflammatory process [51]. People with a vulnerable genetic constitution could have mitochondrial damage caused immunologically and neural oxidative stress [52].

Mattock and colleagues [53] hypothesized that exposure to influenza virus in utero could cause harm to the fetal substantia nigra, hence increasing the risk of Parkinson’s disease in adulthood; however, this discovery was not verified [54]. Fazzini et al. [55] discovered elevated coronavirus antibody titers in the cerebrospinal fluid of PD patients, while Hubble and colleagues [56] and Kohbata and Shimokawa [57] discovered elevated Nocardia antibody titers. Nocardia has a unique preference for substantia nigral cells was shown to cause a levodopa-responsive movement impairment in rats [58], although in a subsequent case-control study, Nocardia titers were not elevated [56]. In a study [59], it was observed that PD patients were more inclined to recall childhood illnesses with diphtheria and croup compared to controls, but there were no variations in the incidence of other diseases.

### 5.2. Parkinson’s Disease and HCV (Hepatitis C Virus)

Nonetheless, other research suggests that infection with certain pathogenic microorganisms can reduce or have no effect on the incidence of PD [60]. Due to the great incidence of various diseases, this infection-related increase in PD risk could impact a sizable portion of the population. HCV was found to infect 0.3% of the population in Spain [61]; HP repopulates the stomach epithelium of almost fifty percent of the world’s population [62]; Malassezia was discovered on more than 80% of lesioned skin [63]; and HBV (hepatitis B virus) affects approximately 360 million chronic carriers and 2 billion individuals globally [64]. Knowing the effect of such infections on the chance of developing Parkinson’s disease is essential, thus helping with the detection and prediction of PD and benefitting a wide population. In addition to enhancing our awareness of the pathophysiology of PD, more research on the process by which these infections raise the risk of Parkinson’s disease might even enhance our knowledge of this condition.

Antiviral treatment for HCV has been found to lower the risk of Parkinson’s disease in individuals with HCV infection, suggesting that therapy against pathogenic microorganisms could be a possible viable technique for preventing PD [65]. Furthermore, it has not been determined if medication against pathogenic bacteria other than HCV reduces the incidence of PD, thus supporting the hypothesis [65].

It has been observed that the hepatitis C virus increases the risk of Parkinson’s disease [66]; a prior meta-analysis similarly showed a higher prevalence of PD in individuals with HCV infection [67]. It has been claimed that HCV causes PD by stimulating the production of inflammatory cytokines and causing damage to dopaminergic neurons [68]. It has been discovered that the critical HCV receptors CD81, claudin-1, occludin, LDLR, and scavenger receptor-B1 are represented on brain microvascular endothelial cells, a significant constituent of the blood–brain barrier, indicating that HCV can invade the CNS via the described receptors [68]. Key HCV receptors were found to be present on brain microvascular endothelial cells, a crucial part of the BBB, which might offer a pathway for HCV to infiltrate the brain and induce neuroinflammation [69]. This notion was supported by the finding of HCV RNA sequences in autopsy neural tissue from people infected [70]. HCV-induced production of inflammatory cytokines can potentially contribute to the development of Parkinson’s disease. In animal studies, 60% of dopaminergic neurons in the rat midbrain perished due to HCV infection.

HCV had a comparable damaging impact on dopaminergic neurons as 1-methyl-4-phenylpyridinium (MPP+) and enhanced the probability of Parkinson’s disease in individuals [68]. One meta-analysis revealed that the risk of Parkinson’s disease in HCV patients who were given efficient antiviral treatment for HCV is significantly smaller compared to those who were not treated with such treatment, suggesting that HCV could be a potential risk for PD and that antiviral therapy for HCV [65] might lower the risk of PD [71].

Consequently, efficient, and more aggressive antiviral therapy must be explored for HCV patients; the relationship among HCV burden and risk of PD requires additional investigation. Interestingly, findings indicated that using interferon-based antiviral treatment for HCV elevated the incidence of PD; this could be owing to an elevation in drug-induced parkinsonism among interferon-treated individuals [65].

### 5.3. Parkinson’s Disease and HP

It was discovered that HP infection increases the production of MPTP [72] or a chemical similar to MPTP and causes persistent inflammation in the central nervous system, which damages dopaminergic neurons [73] by activating microglia [74], producing neurotoxic compounds [75], or promoting autoimmunity [76]. In addition to affecting PD symptoms by decreasing levodopa absorption, HP infection was associated with impaired motor performance in PD patients [77]. Consequently, HP infection could be a possible factor in the beginning of PD. Given the significant incidence of HP infection, it could be appropriate to think about identification and eradication of HP in individuals with a family background with Parkinson’s disease or a high risk for PD. In individuals with Parkinson’s disease, eradication of HP can improve motor symptoms or enhance the impact of levodopa; nevertheless, whether eradication of HP influences the natural process or development of PD requires more study.

### 5.4. Parkinson’s Disease and EBV

Epstein–Barr virus (EBV) seropositivity in PD patients was shown to be greater compared to the general population, according to statistical data. Unusual manifestations of Parkinsonism in EBV infection, especially EBV encephalitis, include akinetic-rigid mutism, tremor, and eyelid opening apraxia. There have been reports of morphological brain abnormalities, involving gradual putaminal and caudate degeneration, with one instance demonstrating a definite acute neutropic action of EBV on substantia nigra dopamine neurons [78].

### 5.5. Parkinson’s Disease and VZV

Herpes zoster is determined by the revival of dormant varicella-zoster virus (VZV) due to a reduction in cell-mediated immunity in humans [79]. A population-based cohort research in Taiwan indicated that patients with a past infection with herpes zoster (age > 65) had an elevated chance of developing Parkinson’s disease [80]. Due to the common pathways of neuroinflammation, immunological alterations, and neuronal loss in both illnesses, herpes zoster individuals might exhibit characteristic Parkinson’s clinical signs during the follow-up duration [80]. Those manifestations, especially during the initial three months following a confirmation of herpes zoster, may result in a prompt PD diagnosis [80].

### 5.6. Associations of the Most Encountered Infections with the Development of Parkinson’s Disease Documented in the Literature

In particular, prior research indicated that infection with pertussis, scarlet fever, HBV, herpes virus, influenza virus, measles, and mumps enhanced the risk of developing Parkinson’s disease [60].

Infections with diphtheria [81], rubella [81], dengue viral infections [82,83,84] and rheumatic fever [81], EBV [85], CMV [85], TB [86], HHV-6 [87], VZV [87], and malassezia [88] have all been linked to an increased risk of PD (Table 1). To elucidate the involvement of these microbial pathogens in PD, further research is required. Nowhere previously have the methods through which these pathogenic microorganisms raise the probability of PD onset been investigated.

Infection with additional pathogenic microbes, such as Coxsackie virus, Japanese encephalitis B virus, West Nile virus, Louis encephalitis virus, HIV, Plasmodium falciparum, enteroviruses, and dengue viral infections, have also been discovered to cause postencephalitic parkinsonism involving bilateral substantia nigra [82,83,84]. It is plausible that infection with these harmful microbes might potentially cause PD via a chronic and progressive mechanism such as persistent neuroinflammation [69]. This shows that infections with pathogenic microorganisms can play complex roles in the etiology of Parkinson’s disease and might even possibly impact tissues in addition to the substantia nigra associated with PD [69,82,83,84]. To better understand how infection contributes to the development of Parkinson’s disease, further study is necessary.

### 5.7. Infectious Risk Variables

Nocardia asteroides is among the bacterial influencing factors that have been investigated in Parkinson’s disease causative agents in the last years, beginning with the research of Kohbata et al. [58] that noticed the evolution of motor irregularities in animal studies colonized with a strain of Nocardia asteroides, which are responsive to levodopa and also presents with inclusions in neurons. Regarding animal models, it has been observed that Nocardiae can disseminate from neuroglia to neurons. These are observed to proliferate inside astroglia, therefore invading midbrain neurons and inducing neuronal degeneration and Lewy body production [89]. Nevertheless, the connection needs to be clarified due to the absence of confirmation for Nocardia asteroides in cerebral samples from patients with diseases that present Lewy bodies [90].

Proteus mirabilis (typically elevated in the gut microbiome of PD animal studies) was already demonstrated to generate PD-correlated pathophysiological alterations and movement impairments. It entails the promotion of alpha-synuclein aggregates in the cerebrum and gut of PD animals as well as the production of dopaminergic neurodegeneration and inflammatory reactions in the substantia nigra and basal ganglia [91].

Lipopolysaccharide (LPS), a P. mirabilis pathogenic agent, was already linked to such inflammatory processes. It is therefore believed that oxidative stress and neuronal pathogenic alpha-synuclein aggregation are caused by higher intestinal permeability that exposes the gut’s neural cells to more pro-inflammatory compounds [91,92].

Many bacteria and viruses, notably Chlamydia pneumoniae [78,93], Bordetella pertussis [94,95], Streptococcus pyogenes [94,96], and Borrelia burgdoferi [94,96], were studied in relation to Parkinson’s disease [78]. There is little data to demonstrate the significance of experimental and epidemiological research in the onset of Parkinson’s disease, and hence the differing findings investigations are largely informative. The association among these bacterial etiological factors and PD will need to be investigated in detail in the long term.

## 6. Reflections and Prospective Viewpoint

In the recent period, substantial proof has implicated viral etiological factors in the neural circuits that contribute to the onset of Parkinson’s disease. Several infections have been shown to be independently associated with Parkinson’s disease in investigations. Furthermore, recognized is the cumulative action of infectious agents in generating inflammatory processes in the brain that leads to the onset of Parkinson’s disease. Neuroinflammation might not just contribute to the onset of Parkinson’s disease but also to its development. The pathophysiology of dopamine neuronal damage in infection-related Parkinsonism may not elucidate the etiology of PD. It is impossible to demonstrate that every instance of PD is connected with elevated inflammatory processes and concomitant chronic infection. This is owing to numerous variables, along with the notion that not every PD patient exhibits consistent proof of inflammatory cytokine imbalance and that a chronic inflammatory status does not necessarily lead to the onset of PD [14,36]. Those with a genetic susceptibility to Parkinson’s disease can generate the condition as a consequence of an existing immune system imbalance [14,36].

## 7. Conclusions

Additional study is required to determine the degree to which infections and inflammatory cytokines contribute to the pathogenesis of Parkinson’s disease. By gaining a deeper knowledge of these systems, we could be capable of classifying variations of Parkinson’s disease, utilizing biomarkers to help in the identification and prediction of therapeutic efficacy and modify therapy to enhance patients’ prognoses. Symptoms of Parkinsonism after viral etiologies, infectious implications for inflammatory processes, and neurodegeneration aspects correlated with Parkinsonism, and the epidemiological link among pathogenic organisms and idiopathic PD are described.

This review examines the association among essential pathogenic organisms and parkinsonism, such as parkinsonism symptoms following infectious etiological factors, infectious contributions to neuroinflammation and neurodegenerative aspects correlated with parkinsonism, and epidemiologic associations among pathogenic organisms and idiopathic PD. There is presently no treatment or disease-modifying medication that can halt or decrease the course of Parkinson’s disease. Nevertheless, a rising number of studies show that anti-infective drugs are especially suitable for the treatment of Parkinson’s disease.

Infection could have a unique role in the development of PD, and anti-infective drugs are possible treatment targets for the treatment of PD according to the findings of recent investigations. This theory requires more experimental and epidemiological investigations to be confirmed.

## Figures and Tables

**Figure 1 life-13-00805-f001:**
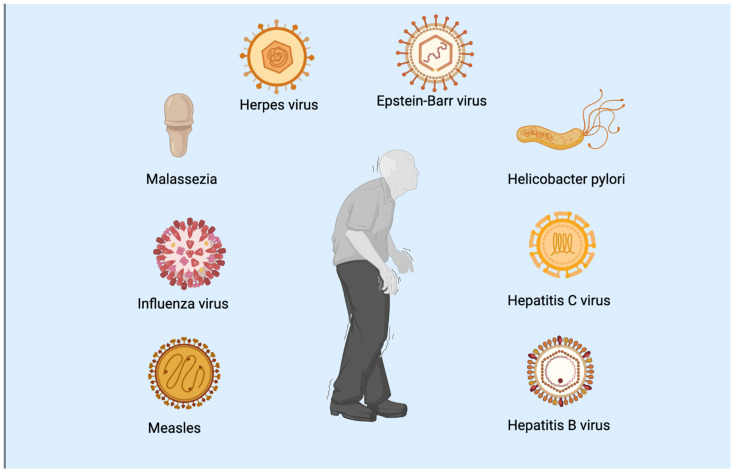
The pathogens implicated in the development of PD.

**Table 1 life-13-00805-t001:** Associations of infections with the development of Parkinson’s disease in the literature.

Infectious Agent	Reference
Hepatitis C virus	[67]
Hepatitis B virus	[67]
Helicobacter pylori	[77]
Epstein–Barr virus	[85]
Chlamydophila pneumoniae	[85]
Cytomegalovirus	[85]
Borrelia burgdorferi	[85]
Influenza virus	[86]
Measles	[86]
Mumps	[86]
German measles	[86]
Pertussis	[86]
Scarlet fever	[86]
Rheumatic fever	[86]
Diphtheria	[86]
Herpes virus	[87]
Varicella–zoster virus	[87]
Malassezia	[88]

## Data Availability

Not available.

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
