# Peer review of "Infectious Microorganisms Seen as Etiologic Agents in Parkinson’s Disease"

_life, 2023, doi:10.3390/life13030805_

Round 1

Reviewer 1 Report

The authors Zorina et al., in the manuscript entitled The infectious microorganisms - seen as etiologic agents in Parkinson's disease

carry out an important review on the infectious- type epigenetic hypothesis on Parkinson's disease.

The authors highlight all the signals left by other authors in the literature where they glimpse the connection between infectious pathogens and neuronal disorders.

The manuscript is very interesting.

I only ask the authors to correct the minor revisions. It is appropriate that at the end of each sentence, they insert the reference.

Minor revision

1. Line 38. Add reference, please

2. Like 42: add a reference, please.

3. Line 74: Please add references and increase the period, and list studies highlighting the involvement of the cholinergic system in Parkinson's disease

4. From line 96 to line 102: to each sentence, add the reference

5. Line 103: the first time, always write the whole word and then use abbreviations: central nervous system ( CNS)

6. Line 113, add a reference

7. Line 169 : like line 103: HLA-DR (Human Leukocyte Antigen – DR isotype)

8. Line 172: like line 103: TNF-alpha…….etc.…..every abbreviation, please.

9. Line 184: add reference, please

10. Line 202,204, 207, 209: add the reference, please

11. Line 212: rural residency; this phrase isn’t really exact. There are many places not rural with the same statistical data of PD. Please re-write the sentence.

12. Line 232: HCV……correct like suggested in line 103

13. Line 234 HBV

14. Lines 254-256: add reference, please

15. Line 286: …correct like suggested in line 103

16. Line 304, add the reference, please

17. Line 325, 326: add references, please

Author Response

Dear Reviewer,

Thank you so much for taking the time to read and analyze our manuscript. We have fulfilled your requirements, and we hope we can bring you a version of the manuscript that will be in accordance with your expectations. We are grateful for your help and kindness.

The authors Zorina et al., in the manuscript entitled The infectious microorganisms - seen as etiologic agents in Parkinson's disease

carry out an important review on the infectious- type epigenetic hypothesis on Parkinson's disease.

The authors highlight all the signals left by other authors in the literature where they glimpse the connection between infectious pathogens and neuronal disorders.

The manuscript is very interesting.

I only ask the authors to correct the minor revisions. It is appropriate that at the end of each sentence, they insert the reference.

Minor revision

  1. Line 38. Add reference, please

Done. We have made the modifications. The lines may not have the same numbering because of the modifications required by the other reviewer.

  1. Like 42: add a reference, please.

Done. We have made the modifications. The lines may not have the same numbering because of the modifications required by the other reviewer.

  1. Line 74: Please add references and increase the period, and list studies highlighting the involvement of the cholinergic system in Parkinson's disease

Done. We have made the modifications. The lines may not have the same numbering because of the modifications required by the other reviewer.

  1. From line 96 to line 102: to each sentence, add the reference

Done. We have made the modifications. The lines may not have the same numbering because of the modifications required by the other reviewer.

  1. Line 103: the first time, always write the whole word and then use abbreviations: central nervous system ( CNS)

Done. We have made the modifications. The lines may not have the same numbering because of the modifications required by the other reviewer.

  1. Line 113, add a reference

Done. We have made the modifications. The lines may not have the same numbering because of the modifications required by the other reviewer.

  1. Line 169 : like line 103: HLA-DR (Human Leukocyte Antigen – DR isotype)

Done. We have made the modifications. The lines may not have the same numbering because of the modifications required by the other reviewer.

  1. Line 172: like line 103: TNF-alpha…….etc.…..every abbreviation, please.

Done. We have made the modifications. The lines may not have the same numbering because of the modifications required by the other reviewer.

  1. Line 184: add reference, please

Done. We have made the modifications. The lines may not have the same numbering because of the modifications required by the other reviewer.

  1. Line 202,204, 207, 209: add the reference, please

Done. We have made the modifications. The lines may not have the same numbering because of the modifications required by the other reviewer.

  1. Line 212: rural residency; this phrase isn’t really exact. There are many places not rural with the same statistical data of PD. Please re-write the sentence.

Done. We have made the modifications. The lines may not have the same numbering because of the modifications required by the other reviewer.

  1. Line 232: HCV……correct like suggested in line 103

Done. We have made the modifications. The lines may not have the same numbering because of the modifications required by the other reviewer.

  1. Line 234 HBV

Done. We have made the modifications. The lines may not have the same numbering because of the modifications required by the other reviewer.

  1. Lines 254-256: add reference, please

Done. We have made the modifications. The lines may not have the same numbering because of the modifications required by the other reviewer.

  1. Line 286: …correct like suggested in line 103

Done. We have made the modifications. The lines may not have the same numbering because of the modifications required by the other reviewer.

  1. Line 304, add the reference, please

Done. We have made the modifications. The lines may not have the same numbering because of the modifications required by the other reviewer.

  1. Line 325, 326: add references, please

Done. We have made the modifications. The lines may not have the same numbering because of the modifications required by the other reviewer.

Reviewer 2 Report

Infections as one of the causes of Parkinson's disease are an extremely interesting topic to review. The content of the manuscript is clear and structured. I have no complaints about the review material, but there are some minor deficiencies related to the presentation. For example, the section on neuroinflammation is divided into many smaller sections, and the section on infections is general for all infections, and section 9 is separately allocated. I suggest to make some general sections, for example, neuroinflammation and then make 3.1, 3.2 in it, where there would be subsections about microglia, astrocytes, etc. The same goes for the section describing infectious agents. Add subsections about each infection, and make section 9 "Miscellaneous Microbial Cause" a subsection under "Infectious risk variables".  

The second comment is about Table 1. This table contains no information, the references are described in the text. I suggest you either remove it or add a paragraph summarizing how the infectious agent is related to Parkinson's disease, a brief summary. 

Otherwise, I really enjoyed the manuscript and found it very interesting. 

Author Response

Dear Reviewer,

Thank you so much for taking the time to read and analyze our manuscript. We have fulfilled your requirements, and we hope we can bring you a version of the manuscript that will be in accordance with your expectations. We are grateful for your help and kindness.

Infections as one of the causes of Parkinson's disease are an extremely interesting topic to review. The content of the manuscript is clear and structured. I have no complaints about the review material, but there are some minor deficiencies related to the presentation. For example, the section on neuroinflammation is divided into many smaller sections, and the section on infections is general for all infections, and section 9 is separately allocated.

I suggest to make some general sections, for example, neuroinflammation and then make 3.1, 3.2 in it, where there would be subsections about microglia, astrocytes, etc.

Done. We have made the modifications.

The same goes for the section describing infectious agents. Add subsections about each infection, and make section 9 "Miscellaneous Microbial Cause" a subsection under "Infectious risk variables".  

Done. We have made the modifications.

The second comment is about Table 1. This table contains no information, the references are described in the text. I suggest you either remove it or add a paragraph summarizing how the infectious agent is related to Parkinson's disease, a brief summary. 

Done. We have made the modifications.

Otherwise, I really enjoyed the manuscript and found it very interesting. 

Thank you very much!
